# Association between childhood adversities and long-term suicidality among South Africans from the results of the South African Stress and Health study: a cross-sectional study

Belinda Bruwer,[1] Ravi Govender,[1] Melanie Bishop,[1] David R Williams,[2,3] Dan J Stein,[4] Soraya Seedat[1]

► Prepublication history and additional material is available. To view please visit the journal (http://dx.doi.org/10.1136/bmjopen-2013-004644).

For numbered affiliations see end of article.

**Correspondence to**
Dr Belinda Bruwer;
bbruwer@sun.ac.za

## ABSTRACT

**Objective:** Suicide and suicidal behaviours are significant public health problems and a leading cause of death worldwide and in South Africa. We examined the association between childhood adversities and suicidal behaviour over the life course.

**Methods:** A national probability sample of 4351 South African adult participants (aged 18 years and older) in the South African Stress and Health (SASH) study was interviewed as part of the World Mental Health Surveys initiative. Respondents provided sociodemographic and diagnostic information, as well as an account of suicide-related thoughts and behaviours. Suicidality or suicidal behaviour were defined as were defined as suicide attempts and suicidal ideation in the total sample, and suicide plans and attempts among ideators. Childhood adversities included physical abuse, sexual abuse, parental death, parental divorce, other parental loss, family violence, physical illness and financial adversity. The association between suicidality and childhood adversities was examined using discrete-time survival models.

**Results:** More than a third of the respondents with suicidal behaviour experienced at least one childhood adversity, with physical abuse, parental death and parental divorce being the most prevalent adversities. Physical abuse, sexual abuse and parental divorce were identified as significant risk markers for lifetime suicide attempts, while physical abuse and parental divorce were significantly correlated with suicidal ideation. Two or more childhood adversities were associated with a twofold higher risk of lifetime suicide attempts. Sexual abuse (OR 9.3), parental divorce (OR 3.1) and childhood physical abuse (OR 2.2) had the strongest associations with lifetime suicide attempts. The effect of childhood adversities on suicidal tendencies varied over the *life course.* For example, sexual abuse was significantly associated with suicide attempts during childhood and teen years, but not during young and later adulthood.

**Conclusions:** Childhood adversities, especially sexual abuse, physical abuse and parental divorce, are important risk factors for the onset and persistence of suicidal behaviour, with this risk being greatest in childhood and adolescence.

### Strengths and limitations of this study

■ Recall bias might have impacted on the accuracy of recall of childhood adversities.
■ In view of the cross-sectional design, more detailed, temporal information on childhood adversities and suicidal incidents was not obtained.
■ Variables such as culture, ethnicity and mental status at the time of the interview may have influenced the recall and reporting of suicidal behaviour. Stigma associated with mental health problems may have also played a role in reporting suicidal tendencies.
■ We did not assess for self-mutilating behaviour. The importance of discriminating suicidal behaviour from non-suicidal self-mutilation cannot be underestimated. The Composite International Diagnostic Interview (CIDI) instrument which was used in this study is a lay-administered instrument which does not include an assessment of several key Diagnostic and Statistical Manual of Mental Disorders (DSM)-IV diagnoses (such as bipolar disorder and psychosis), which are associated with elevated rates of suicidality. As a result, some participants with suicidality may not have been diagnosed with a disorder.
■ This study represents the first investigation among South Africans of a wide range of childhood adversities and their impact on the onset and persistence of suicidality over the life course.

## INTRODUCTION

Suicide and suicidal behaviour are significant public health problems. Suicide is one of the leading causes of death worldwide with almost one million people committing

suicide each year.[1] This figure is likely to grow to approximately 1.2 million suicides in 2020.[2] In South Africa, the annual rate of suicide is high,[3] [4] mirroring international trends.[5] So, too, are rates of suicidal behaviour with an estimated prevalence of 9.1% for lifetime suicidal ideation and 2.9% for suicide attempts among South Africans according to the South African Stress and Health (SASH) study.[6]

Despite the enormity of the problem, the aetiology of suicidal behaviour is not fully understood. There are controversies in the literature regarding prior psychiatric disorder and risk for suicide attempts. While some authors have argued that pre-existing disorder is an important risk factor,[7–11] others have argued that suicide attempts are not neccessarily associated with prior psychopathology.[12] Genetic factors also play an important role in suicidal behaviour.[13–16] While there is stronger evidence pointing towards environmental or experiential factors[17] [18] such as exposure to childhood adversities.[19–28] Recent multilevel country data from the World Mental Health Surveys (WMHS) initiative has allowed for cross-national comparisons of suicidality. The WMHS investigated the association between childhood adversities and suicidal behaviour,[20] the persistence of suicidality over time and the extent to which associations between childhood trauma and suicidality changed over the life course. The WMHS found a dose–response relationship between the number of adversities and suicidal behaviour. Sexual abuse and physical abuse were the strongest risk factors for the onset and persistence of suicidal behaviours, with the risk for suicidality being greatest during childhood (age 4–12 years) and adolescence (age 13–19 years).[20]

Numerous studies have examined the link between childhood sexual abuse and suicidality.[29–41] All of these authors have found that exposure to childhood sexual abuse increases the risk for mental disorders, including suicidality. Furthermore, the majority of studies that have focused on the link between childhood physical abuse and suicidality have found that exposure to childhood physical abuse increases the risk for suicidality.[42] [43] There also appears to be an association between the number of childhood adversities experienced and the later suicidal behaviour.[21 23 24 44 45]

Exposure to early life stress is prevalent among South Africans. In one sample of South African rural youth, the prevalence of physical and sexual abuse was shown to be very high with 94.4% of men exposed to physical abuse and 39.1% of women to sexual abuse.[46] More than a quarter of the adults who were interviewed endorsed exposure to childhood adversity (parental death, parental separation or parental divorce) in the SASH study.[47] Significantly more women were prone to be victims of domestic violence than men.[47] Women also reported twice as many suicidal attempts as the male participants in the SASH study.[9]

## Objective

We report in more detail on data from a South African dataset gathered as part of the WMHS, which allowed for comparison with data from the overall cross-national sample. These data are particularly interesting as South Africa is a middle-income African country with high rates of violent trauma exposure. The present study aimed to examine the relationship between the type and frequency of childhood adversity exposure and suicidal behaviour over the life trajectory of South Africans, given that there are no published nationally representative data that may be useful in informing clinical practice and policy.

## METHODS
### Sample
Data for the SASH study were collected between January 2002 and June 2004. WMHS were carried out in 21 countries which included Nigeria and South Africa.[48] For detailed information on study methods see Williams *et al*.[48] The study was a national probability sample of 4351 South African adults (persons aged 18 years and older) living in households or in hostel accommodation. All racial and ethnic groups were represented, with the sample selected using a three-stage probability sample design. The response rate was 85.5%.

### Sampling approach
Sampling was divided into three stages. Primary sampling units were selected during the first stage, which was based on the 2001 South African census enumeration areas (EAs). The second stage involved sampling of household units within clusters selected in each EA. South Africans in urban and rural areas were sampled. Sampled residences were stratified into 10 diverse housing categories: rural–commercial, agricultural, rural traditional subsistence areas, African townships, informal urban or periurban shack areas, coloured townships, Indian townships, general metropolitan residential areas, general large metropolitan residential areas and domestic servant accommodation in urban areas. During the third stage, one adult respondent in each sampled housing unit was selected. A total of 5089 households were selected. Field interviews were conducted with 4433 (87.1%) designated respondents. Based on quality control, 4351 interviews were retained for use in the analysis. There were no differences in response rates across the four designated racial groups (Caucasian, coloured (mixed racial origin), Indian, African-American). According to the 2001 census statistics, 79% of the people in South Africa are black African, 8.9% are coloured, 9.6% are Caucasian and 2.5% are Indian/Asian.[49]

## Diagnostic interview

SASH used V.3 of the WHO Composite International Diagnostic Interview (WHO CIDI).[50] Interviewers were trained within a 1-week period and conducted the interviews in seven different languages, namely English, Afrikaans, Zulu, Xhosa, Northern Sotho, Southern Sotho and Tswana. Translations of the CIDI into several native South African languages were conducted in accordance with WHO requirements. Multilingual and bilingual expert panels conducted the back translations.[51 52] Informed consent was obtained from participants after a complete description of the study was provided. Respondents provided sociodemographic and diagnostic information, as well as an account of suicidal behaviours during the interviews. The core diagnostic assessment of mental disorders included anxiety disorders (panic disorder, agoraphobia, social phobia, generalised anxiety disorder, post-traumatic stress disorder), mood disorders (major depressive disorder, dysthymia), substance use disorders (alcohol abuse, alcohol dependence, drug misuse, drug dependence) and intermittent explosive disorder.[53 54]

## Suicidal behaviour

The CIDI 3.0 module on suicidal behaviour was used to assess the age of first onset, age of most recent episode, lifetime occurrence of suicidal ideation, suicide plans and suicide attempts. Suicidal ideation, suicide plans and suicide attempts were assessed with questions such as "Have you ever seriously thought about committing suicide?" "Have you ever made a plan for committing suicide?" and "Have you ever attempted suicide?", respectively. Ideators only proceeded to answer questions about plans ("Have you ever made a plan for committing suicide?") and attempts ("Have you ever attempted suicide?"). Information on the age of first occurrence of the three main outcomes was obtained. To get a better understanding of the progression from ideation to attempt, the outcomes considered in this study were: suicide attempts in the total sample, suicide ideation in the total sample, suicide plans among ideators, suicide attempts among ideators with a plan (planned attempts) and suicide attempts among ideators in the absence of a plan (unplanned or impulsive attempts).

## Childhood adversities

Physical abuse, sexual abuse, parental death, parental divorce, other parental loss, family violence, physical illness and financial adversity were the various childhood adversities assessed. Biological and non-biological parents were included in measures of parental death, divorce or other parental loss. Financial adversities were assessed with questions on whether the family had insufficient funds to pay for basic necessities. Questions about repeated fondling, attempted rape or rape were asked to assess for sexual abuse. This comprised the following "The next 2 questions are about sexual assault: (i) The first is about rape. We define this as someone either having sexual intercourse with you or penetrating your body with a finger or object when you did not want them to, either by threatening you or using force, or when you were so young that you didn't know what was happening. Did this ever happen to you?," and (ii) "Other than rape, were you ever sexually assaulted or molested?" A modified version of the Conflict Tactics Scale (CTS2) was used to assess family violence and physical abuse.[55] Respondents were classified as having experienced *physical abuse* when they indicated that, when they were growing up, their father or mother (includes biological, step or adoptive parents) slapped, hit, pushed, grabbed, shoved or threw something at them, or that they were beaten as a child by the persons who raised them. Family violence was assessed as present when respondents indicated that they (1) "were often hit, shoved, pushed, grabbed, or slapped while growing up" or (2) "witnessed physical fights at home, like when your father beat up your mother?" A standard chronic conditions checklist assessed life-threatening physical illnesses in childhood.[56]

## Data analysis

All data analyses were processed and analysed centrally by a team of statisticians at the Harvard School of Public Health (Boston, USA) using the SAS V.9.1.3 software package. Discrete-time survival analysis with time-varying covariates was used to study the risk factors of lifetime suicide ideation, plans and attempts. Data were weighted to adjust for the stratified multistage sample design, differential probability of selection within households as a function of household size and clustering of data and differential non-response. Overall, percentages were weighted to adjust for differences in selection probabilities, differential non-response, oversampling of cases and residual differences on sociodemographic variables between the sample and the population.[48 57] A poststratification weight was also used to make the sample distribution comparable, for age, sex and province, with the population distribution in the 2001 South African census. Weighted and geographic clustering of data were taken into account in the data analyses by using a jackknife repeated replications simulation method implemented in SAS macro V.14. The survival coefficients were exponentiated and are reported below in the form of ORs.

The association between suicidality and childhood adversity was examined using discrete-time survival models with the analysis unit being person-years. Bivariate analyses (considering one adversity at a time) and multivariate analyses (considering all adversities simultaneously) were conducted. Two types of multivariate models were tested: multivariate additive models (simultaneously considering all childhood adversities) and multivariate interactive models (with number and type of childhood adversities experienced by each respondent included as dummy variables). The analysis also examined interactions between the life stage (13–19,

20–29, 30+ years) of respondents and each childhood adversity, as well as the influence each adversity had on early-onset, middle-onset and later-onset suicidality. Analyses were conducted using SUDAAN V.8.1 to adjust for clustering and weighting. ORs with a 95% CI are reported. Wald $X^2$-tests were used to examine multivariate significance. Associations between adversities and suicide outcomes were adjusted for sex, age, educational level, marital status, interactions between demographic variables, life course, lifetime mental disorders and parental psychopathology. Analyses also examined the influence of respondents' lifetime mental disorders on suicidality, as well as interactions between sex and each childhood adversity. Statistical significance using two-sided tests was set at p<0.05.[20] Based on an N of 4000 (α of 0.05, two-sided significance), the study was adequately powered (0.99) to detect an OR of 2.0 of a continuously distributed normalised predictor and a 10% prevalence of suicidal behaviour.

## RESULTS
### Demographic details
In the sample (n=4351), there were slightly more female (53.7%) than male respondents. Most of the respondents were black (76.2%), followed by coloured (10.4%), white (10%), and Indian/Asian (3.4%). Furthermore, half of the sample was married and most were unemployed (69.2%), had less than 12 years of education (62.7%) and lived in an urban area (59.7%; see table 1).

### Prevalence of childhood adversities among the total sample
Figure 1 provides a schematic representation of the suicidality data reported in the sections which follow. In the total sample, 35.4% of participants with one adversity had a suicide attempt, compared with 23.4% with one adversity who had not made an attempt. Physical abuse (24.9%), parental divorce (14.2%) and parental death (11.6%) were most prevalent among those suicide attempters. Among those exposed to one childhood adversity, without a suicide attempt, the two most prevalent adversities reported were physical abuse (12.2%) and parental death (11.3%). In the total sample, 15.4% of participants exposed to two or more adversities had a suicide attempt. In contrast, 8.6% of participants exposed to two or more adversities had not made an attempt (table 2).

### Prevalence of childhood adversities among suicidal ideators in the total sample
In the sample as a whole, 35.9% of those with one adversity had suicidal ideation compared with 22.7% of those with one adversity who had no ideation. The most prevalent adversities associated with suicidal ideation were physical abuse (21.1%), parental death (13.9%) and parental divorce (7.9%). Among those without suicidal ideation, physical abuse (11.8%) and parental death

**Table 1** Descriptive characteristics (N=4351)

| | | N |
|---|---|---|
| Mean age (years; SE) | 37.0 (0.26) | |
| Age categories (years) | | |
| 18–29 | 39.1% | 1701 |
| 30–39 | 22.1% | 962 |
| 40–49 | 18.1% | 788 |
| ≥50 | 20.7% | 901 |
| Sex | | |
| Male | 46.3% | 2015 |
| Female | 53.7% | 2336 |
| Race | | |
| Black | 76.2% | 3315 |
| Coloured (mixed race) | 10.4% | 453 |
| White | 10.0% | 435 |
| Indian/Asian | 3.4% | 148 |
| Married | 50.1% | 2180 |
| Location | | |
| Rural | 38.4% | 1671 |
| Urban | 61.6% | 2680 |
| Education | | |
| None | 6.8% | 296 |
| Grades 1–7 | 19.1% | 831 |
| Grades 8–11 | 35.4% | 1540 |
| Matric | 23.5% | 1022 |
| Matric+ | 15.3% | 665.7 |
| Employed | 31.0% | 1349 |
| Income category (Rands; mean SD) | | |
| 0 | 13.7% | 596 |
| 1–2500 | 29.5% | 1284 |
| 2501–5000 | 15.4% | 670 |
| 5001–10 000 | 19.6% | 853 |
| ≥10 001 | 21.8% | 949 |
| Province | | |
| Eastern Cape | 13.1% | 570 |
| Free State | 6.2% | 270 |
| Gauteng | 23.0% | 1001 |
| Kwazulu-Natal | 19.5% | 848 |
| Limpopo | 10.5% | 457 |
| Mpumalanga | 6.6% | 287 |
| Northern Cape | 1.9% | 83 |
| North West | 8.3% | 361 |
| Western Cape | 11.1% | 483 |

(11.3%) were the most commonly endorsed childhood adversities. Of those who endorsed two or more childhood adversities, 10.8% reported suicidal ideation and 8.6% did not (table 2). In summary, the most prevalent childhood adversities reported among the total sample with/without suicidal ideation were, first, physical abuse and, second, the death of a parent.

### Prevalence of suicide attempts in the total sample
In the total sample, 24.9% of those with childhood physical abuse had attempted suicide while 12.2% of respondents with no physical abuse had no attempt. Of those exposed to parental divorce, 14.2% had attempted suicide and 4.8% had made no attempt. The second

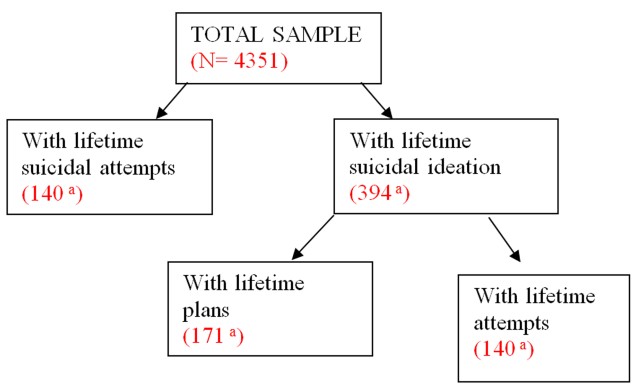

a Number of cases with the outcome variable; N represents the number of person years.

**Figure 1** Schematic representation. [a]Number of cases with the outcome variable; N represents the number of person-years.

most prevalent childhood adversity was parental death with 11.6% of those with parental death attempting suicide and 11.3% of those with parental death with no attempts (table 2).

## Prevalence of childhood adversities among suicidal ideators
### With/without a plan
Among suicidal ideators with a plan, 32.9% had experienced one childhood adversity. Among ideators with no plan, 41.7% had one childhood adversity. Among ideators with a plan, the following were the most prevalent childhood adversities: physical abuse (24.3%), parental death (12.2%) and parental divorce (9.7%). Among ideators without a plan, 27.9% endorsed physical abuse, 16.1% parental death and 9.2% parental divorce (see table 2). In both groups (ideators with and without a plan), physical abuse was the most prevalent childhood adversity, followed by parental death and parental divorce.

### With/without an attempt
Among suicidal ideators who had attempted suicide, 35.4% were exposed to one childhood adversity and 15.4% were exposed to two or more childhood adversities. In the group of ideators who had made an attempt, 24.9% had experienced physical abuse, 14.2% parental divorce and 11.6% parental death (table 2). In total, 40.5% of those with one adversity, and 9.6% of those exposed to two or more adversities were suicidal ideators with no attempts. In this group, the most prevalent adversities reported were physical abuse (24.5%), parental death (15.6%) and parental divorce (6.7%) (table 2).

Among all ideators (with/without a plan, with/without an attempt), the most prevalent childhood adversity was physical abuse, followed by parental death and parental divorce. Of note, in the group of ideators with an attempted suicide, parental divorce was more prevalent than parental death.

## Bivariate and multivariate results: type of childhood adversity
Bivariate and multivariate analyses were performed to examine the associations between the different childhood adversities (physical abuse, sexual abuse, parental death, parental divorce, other parental loss, family violence, physical illness, financial adversity) and lifetime suicidal ideation, plans and attempts.

In the total sample, bivariate and multivariate analysis revealed significant associations between (1) sexual abuse (bivariate: OR 7.9, p=0.003; multivariate: OR 7.6, p=0.003), (2) physical abuse (OR 2, p=0.007; OR 2.0, p=0.006) and (3) parental divorce (OR 2.8, p<0.001; OR 2.7, p=0.001), and lifetime suicide attempts. Among ideators in the sample, physical abuse (OR 1.7, p<0.001; OR 1.7, p<0.001) was significantly associated with suicidal ideation. Multivariate analyses revealed an additional association with suicidal ideation, namely parental divorce (OR 1.6, p=0.038). The relationship

**Table 2** Prevalence of childhood adversities and suicidal behaviour in South Africa (% (SE))

|  | Total sample | | Total sample | | Suicidal ideators | | Suicidal ideators | |
| --- | --- | --- | --- | --- | --- | --- | --- | --- |
|  | With attempt | No attempt | With ideation | No ideation | With plan | No plan | With attempt | No attempt |
| Physical abuse | 24.9 (4.6) | 12.2 (0.8) | 21.1 (2.5) | 11.8 (0.7) | 24.3 (4.6) | 27.9 (4.0) | 24.9 (4.6) | 24.5 (3.6) |
| Sexual abuse | 2.1 (1.2) | 0.1 (0.0) | 0.7 (0.4) | 0.1 (0.0) | 1.6 (0.9) | 0.0 (0.0) | 2.1 (1.2) | 0.0 (0.0) |
| Parent died | 11.6 (2.4) | 11.3 (0.6) | 13.9 (2.3) | 11.3 (0.6) | 12.2 (2.4) | 16.1 (4.2) | 11.6 (2.4) | 15.6 (3.8) |
| Parent divorced | 14.2 (3.8) | 4.8 (0.4) | 7.9 (1.6) | 4.7 (0.4) | 9.7 (2.6) | 9.2 (3.7) | 14.2 (3.8) | 6.7 (2.9) |
| Other parent loss | 2.1 (1.2) | 2.2 (0.4) | 3.9 (1.2) | 2.1 (0.4) | 1.1 (0.6) | 3.0 (1.4) | 2.1 (1.2) | 2.7 (1.3) |
| Family violence | 4.3 (1.5) | 3.0 (0.3) | 4.1 (0.9) | 2.9 (0.3) | 4.7 (1.5) | 6.3 (1.8) | 4.3 (1.5) | 4.5 (1.4) |
| Physical illness | 5.0 (2.3) | 2.5 (0.3) | 4.0 (1.2) | 2.4 (0.3) | 4.4 (1.8) | 4.7 (1.8) | 5.0 (2.3) | 4.3 (1.6) |
| Financial adversity | 6.1 (2.4) | 5.6 (0.5) | 4.1 (0.9) | 5.8 (0.5) | 6.0 (2.1) | 3.3 (1.5) | 6.1 (2.4) | 2.9 (1.0) |
| 1 | 35.4 (4.2) | 23.4 (1.0) | 35.9 (2.8) | 22.7 (0.9) | 32.9 (4.0) | 41.7 (5.2) | 35.4 (4.2) | 40.5 (4.5) |
| 2+ | 15.4 (3.4) | 8.6 (0.5) | 10.8 (1.7) | 8.6 (0.5) | 14.1 (3.2) | 13.2 (3.3) | 15.4 (3.4) | 9.6 (2.3) |
| * | (140) | (107 309) | (394) | (112 243) | (171) | (1976) | (140) | (2212) |

*Number of cases with the outcome variable.
%, the percentage of people with the adversity among cases with the outcome variable indicated in the column header. For example, the first cell is the percentage of those with physical abuse among those with attempts.

between childhood adversities and lifetime plans was not statistically significant. However, a significant association was found between parental divorce and lifetime suicidal attempts among ideators (OR 3.0, p<0.001; OR 3.1, p=0.023; table 3).

Findings from multivariate analysis, therefore, confirm findings of bivariate analysis for all groups, except for the ideators. Among ideators, bivariate analysis revealed a significant relationship between physical abuse and suicidal ideation. This was confirmed in multivariate analysis where the association between parental divorce and suicidal ideation was significant for the whole sample.

## Bivariate associations between the number of adversities and lifetime suicidality

The relationship between the number of childhood adversities and lifetime suicidal ideation, plans and attempts was further examined. There was a significant relationship between the number of childhood adversities and lifetime suicide attempts. Two or more childhood adversities were associated with a twofold higher risk of lifetime suicide attempts in the total sample (OR 2.1, p<0.001). A significant relationship was also established between one, as well as two or more adversities with ideators in the total sample. Among ideators, no significant association was found between the number of childhood adversities and lifetime plans. A significant relationship was found between two or more adversities and lifetime attempts among ideators (OR 2.7, p=0.016), indicating a more than twofold higher risk of lifetime suicide attempts in this group (table 4).

## Multivariate associations between number of childhood adversities and lifetime suicidality

In the final multivariate model which included two or more adversities as a predictor variable, sexual abuse (OR 9.3, p<0.001), childhood physical abuse (OR 2.2, p=0.003) and parental divorce (OR 3.1, p<0.001) retained significant associations with lifetime suicide attempts in the total sample. Physical abuse (OR 2.1, p<0.001), parental death (OR 1.7, p=0.010), parental divorce (OR 1.9, p=0.004) and other parental loss (OR 2.1, p=0.004) were significant predictors of suicidal ideation (table 5). The same findings emerged after controlling for mental disorders, with the exception that sexual abuse was also significantly associated with suicidal ideation (table 6). Physical abuse was associated with a lower OR of lifetime suicide plans among ideators (OR 0.4, p=0.038; table 5). There were no significant associations between childhood adversities and lifetime attempts among those with suicidal ideation. The findings remain unchanged after controlling for mental disorders (table 6).

## Associations between the types of childhood adversity and lifetime suicidality over the life course

Multivariate analyses were performed to examine the association between the types of childhood adversity and lifetime suicidal ideation, plans and attempts during

childhood years (age 4–12), teen years (age 13–19), young adulthood (age 20–29) and later adulthood (30 years and older; see online supplementary table 1).

## Childhood years (4–12)

Sexual abuse (OR 61.6, CI 4.5 to 841.0, p=0.002) in early childhood (4–12 years of age) was significantly associated with lifetime suicide attempts in the total sample (OR 61.6, CI 4.5 to 841.0, p=0.002). Sexual abuse (OR 34.8, CI 3.1 to 392.6, p=0.003) and physical abuse (OR 3.7, CI 1.0 to 13.4, p=0.041) were associated with a higher risk for suicidal ideation among the total sample. No significant associations were found between any of the childhood adversities and lifetime plans in the group of ideators. Among those with suicidal ideation, parental death (OR 2.2, CI 1.1 to 4.3, p=0.021) was significantly associated with suicide attempts in childhood years.

## Teen years (13–19)

Sexual abuse (OR 20.3, CI 2.0 to 210.2, p=0.010), physical abuse (OR 3.7, CI 1.5 to 9.2, p=0.004) and parental divorce (OR 4.6, CI 1.7 to 12.1, p=0.002) were significantly associated with suicide attempts in the total sample of teenagers. Physical abuse (OR 3.6, CI 2.2 to 5.9, p<0.001) and parental death (OR 2.2, CI 1.1 to 4.3, p=0.021) significantly increased the risk for suicidal ideation among the total group of teens. Physical illness (OR 9.9, CI 1.8 to 54.0, p=0.007) significantly increased the risk of suicidal plans in teens with suicidal ideation. Suicide attempts among teens with suicidal ideation was significantly predicted by parental divorce (OR 4.3, CI 1.1 to 17.0, p=0.035).

## Young adulthood (20–29)

None of the childhood adversities were significantly associated with lifetime suicide attempts during young adulthood in the sample overall. An explanation could be that suicide attempts spike earlier and later in life among South Africans, contributing to the lack of significance. Parental loss other than parental death was significantly associated with suicidal ideation (OR 2.9, CI 1.2 to 7.4, p=0.019).

## Later adulthood (≥30)

Childhood physical abuse (OR 2.2, CI 1.0 to 4.8, p=0.035) was significantly predictive of suicidal attempts. The likelihood of suicidal ideation increased significantly in later adulthood if parental loss other than parental death (OR 5.1, CI 2.1 to 12.1, p<0.001) or physical illness had been present during childhood (OR 4.3, CI 1.1 to 15.9, p=0.028). No significant relationship was found between any of the childhood adversities and lifetime plans in the group of ideators, although a significant relationship was found between two or more adversities and lifetime plans among those who were ideators (OR 44.5, CI 2.5 to 779.1, p<0.008). None of

**Table 3** Multivariate and bivariate models for associations between childhood adversities and lifetime (LT) suicidality*

| | LT attempts in total sample† | | | | Ideators among total sample† | | | | Suicidal ideators with LT plans‡ | | | | Suicidal ideators with LT attempts§ | | | |
| | Multivariate | | Bivariate | | Multivariate | | Bivariate | | Multivariate | | Bivariate | | Multivariate | | Bivariate | |
| | OR (95% CI) | χ² | OR (95% CI) | χ² | OR (95% CI) | χ² | OR (95% CI) | χ² | OR (95% CI) | χ² | OR (95% CI) | χ² | OR (95% CI) | χ² | OR (95% CI) | χ² |
|---|---|---|---|---|---|---|---|---|---|---|---|---|---|---|---|---|
| Physical abuse | 2.0 (1.2 to 3.3)¶ | 7.4 (0.006)¶ | 2.0 (1.2 to 3.2)¶ | 7.3 (0.007)¶ | 1.7 (1.3 to 2.3)¶ | 15.2 (<0.001)¶ | 1.7 (1.3 to 2.3)¶ | 16.7 (<0.001)¶ | 0.6 (0.3 to 1.4) | 1.3 (0.25) | 0.7 (0.3 to 1.4) | 1.2 (0.26) | 1.0 (0.5 to 2.3) | 0.0 (0.93) | 1.1 (0.5 to 2.5) | 0.1 (0.81) |
| Sexual abuse | 7.6 (2.0 to 29.9)¶ | 8.9 (0.003)¶ | 7.9 (1.9 to 32.1)¶ | 8.6 (0.003)¶ | 2.6 (0.6 to 10.6) | 1.8 (0.18) | 3.0 (0.7 to 12.2) | 2.5 (0.11) | – | – | – | – | – | – | – | – |
| Parent died | 1.1 (0.6 to 1.8) | 0.1 (0.78) | 1.1 (0.7 to 1.7) | 0.1 (0.76) | 1.4 (0.9 to 2.1) | 2.7 (0.10) | 1.3 (0.9 to 1.9) | 2.0 (0.16) | 0.7 (0.3 to 1.7) | 0.6 (0.45) | 0.8 (0.4 to 1.9) | 0.3 (0.62) | 0.8 (0.5 to 1.5) | 0.4 (0.52) | 0.8 (0.4 to 1.5) | 0.4 (0.53) |
| Parent divorced | 2.7 (1.5 to 5.0)¶ | 10.8 (0.001)¶ | 2.8 (1.5 to 5.2)¶ | 11.4 (<0.001)¶ | 1.6 (1.0 to 2.4)¶ | 4.3 (0.038)¶ | 1.5 (1.0 to 2.3) | 3.7 (0.05) | 0.9 (0.3 to 3.3) | 0.0 (0.88) | 1.2 (0.4 to 3.8) | 0.1 (0.78) | 3.1 (1.2 to 8.6)¶ | 5.2 (0.023)¶ | 3.0 (1.1 to 8.0)¶ | 4.9 (0.027)¶ |
| Other parent loss | 1.0 (0.3 to 3.3) | 0.0 (0.95) | 0.9 (0.3 to 2.8) | 0.1 (0.81) | 1.7 (1.0 to 3.0) | 3.6 (0.06) | 1.6 (0.9 to 2.7) | 2.9 (0.09) | 0.4 (0.1 to 2.6) | 0.9 (0.34) | 0.5 (0.1 to 2.7) | 0.7 (0.41) | 2.0 (0.2 to 17.3) | 0.4 (0.51) | 2.5 (0.6 to 11.0) | 1.5 (0.22) |
| Family violence | 0.7 (0.3 to 1.7) | 0.6 (0.42) | 1.0 (0.4 to 2.2) | 0.0 (0.98) | 0.8 (0.5 to 1.4) | 0.5 (0.47) | 1.1 (0.6 to 1.8) | 0.0 (0.83) | 1.0 (0.4 to 2.4) | 0.0 (0.97) | 0.8 (0.4 to 2.0) | 0.2 (0.68) | 2.4 (0.9 to 6.3) | 3.5 (0.06) | 2.2 (0.9 to 5.5) | 2.9 (0.09) |
| Physical illness | 1.1 (0.4 to 3.5) | 0.1 (0.81) | 1.5 (0.6 to 4.1) | 0.7 (0.39) | 1.2 (0.6 to 2.3) | 0.2 (0.63) | 1.3 (0.7 to 2.4) | 0.7 (0.42) | 0.8 (0.2 to 3.1) | 0.1 (0.71) | 0.9 (0.2 to 3.5) | 0.0 (0.86) | 1.2 (0.3 to 3.9) | 0.1 (0.80) | 1.2 (0.4 to 4.0) | 0.1 (0.77) |
| Financial adversity | 1.0 (0.4 to 2.7) | 0.0 (0.94) | 1.2 (0.5 to 2.8) | 0.1 (0.73) | 0.6 (0.4 to 1.1) | 3.0 (0.08) | 0.7 (0.4 to 1.2) | 1.4 (0.23) | 2.4 (0.7 to 8.4) | 1.9 (0.17) | 1.9 (0.6 to 6.8) | 1.1 (0.29) | 2.1 (0.7 to 6.0) | 2.1 (0.15) | 2.0 (0.7 to 6.3) | 1.6 (0.21) |
| Group significant test of controls: demographic variables | 403.8 (<0.001)¶ | | | | 1102.1 (<0.001)¶ | | | | 12.0 (0.002)¶ | | | | | | | |
| Group significance of controls: interactions between demographics and intervals | 816.6 (<0.001)¶ | | | | 1374.8 (<0.001)¶ | | | | 159.9 (<0.001)¶ | | | | | | | |
| Group significance of controls: demographics and interactions between demographics and intervals | 8369.9 (<0.001)¶ | | | | 6190.0 (<0.001)¶ | | | | 529.0 (<0.001)¶ | | | | | | | |
| Group significance of controls: parent psychopathology | 12.0 (0.003)* | | | | 16.7 (<0.001)* | | | | | | | | | | | |

*Assessed in part 2 sample due to having part 2 controls. Controls for the model include intervals (1–5 intervals), and also include significant variables from demographic and parent psychopathology models, details in the following footnotes.
†Model controls for intervals (1–5 intervals), demographics (sex, age, time-varying education), interaction between intervals (13–19, 20–29, 30+) and age, education. For parent psychopathology, controlling for number of parental disorders (dummies for 1, 2+ disorders).
‡Model controls for intervals (1–5 intervals), demographics (sex, age, time-varying education), interaction between intervals (13–19, 20–29, 30+) and age, education. For parent psychopathology, controlling for types of parental disorders (6 dummies).
§Model controls for intervals (1–5 intervals), demographics (sex, age, time-varying education), interaction between intervals (13–19, 20–29, 30+) and age, education. Parent psychopathology not controlled for due to insignificance in previous models.
¶Significant at the 0.05 level, two-sided test.

**Table 4** Associations between number of childhood adversities and lifetime (LT) suicidality*

| Number of child adversities | LT attempts in total sample† OR (95% CI) | χ² | Ideators among total sample† OR (95% CI) | χ² | Ideators with LT plans‡ OR (95% CI) | χ² | Ideators with LT attempts§ OR (95% CI) | χ² |
|---|---|---|---|---|---|---|---|---|
| 1 | 1.9 (1.3 to 2.8)¶ | | 1.8 (1.5 to 2.3)¶ | | 0.5 (0.3 to 1.0) | | 0.9 (0.5 to 1.7) | |
| 2+ | 2.1 (1.2 to 3.8)¶ | | 1.4 (1.0 to 2.0)¶ | | 1.1 (0.3 to 3.3) | | 2.7 (1.3 to 5.9)¶ | |
| Group significant test of controls: demographic variables | | 14.3 (<0.001)¶ | | 28.3 (<0.001)¶ | | 4.5 (0.10) | | 8.3 (0.016)¶ |
| Group significance of controls: interactions between demographics and intervals | | 538.4 (<0.001)¶ | | 1146.6 (<0.001)¶ | | 600.3 (<0.001)¶ | | 1943.4 (<0.001)¶ |
| Group significance of controls: demographics and interactions between demographics and intervals | | 859.8 (<0.001)¶ | | 1473.5 (<0.001)¶ | | 1389.0 (<0.001)¶ | | 1657.7 (<0.001)¶ |
| Group significance of controls: demographics | | 11 496.7 (<0.001)¶ | | 7255.3 (<0.001)¶ | | 11 233.3 (<0.001)¶ | | 6714.6(<0.001)¶ |
| Group significance of controls: parent psychopathology | | 12.7 (0.002)¶ | | 19.1 (<0.001)¶ | | 14.1 (0.029)¶ | | |

*Assessed in part 2 sample due to having part 2 controls. Controls for the model include intervals (1–5 intervals), and also include significant variables from demographic and parent psychopathology models, details in the following footnotes.
†Model controls for intervals (1–5 intervals), demographics (sex, age, time-varying education), interaction between intervals (13–19, 20–29, 30+) and age, education. For parent psychopathology, controlling for number of parental disorders (dummies for 1, 2+ disorders).
‡Model controls for intervals (1–5 intervals), demographics (sex, age, time-varying education), interaction between intervals (13–19, 20–29, 30+) and age, education. For parent psychopathology, controlling for types of parental disorders (6 dummies).
§Model controls for intervals (1–5 intervals), demographics (sex, age, time-varying education), interaction between intervals (13–19, 20–29, 30+) and age, education. Parent psychopathology not controlled for due to insignificance in previous models.
¶Significant at the 0.05 level, two-sided test.

the childhood adversities were significantly associated with suicide attempts among ideators in this age group.

## DISCUSSION

Rates of childhood adversities and suicidal behaviours were high among South Africans, with more than a third of the respondents in the total sample who attempted suicide experiencing one childhood adversity, and 15.4% experiencing two or more adversities. Overall, physical abuse, sexual abuse, parental divorce and physical illness were far more prevalent in those with a suicide attempt than in those without. The most prevalent childhood adversities endorsed overall were physical abuse followed by parental death. Physical abuse, parental divorce and death of a parent were also the most prevalent adversities experienced in those with a suicide attempt as well as in those with suicidal ideation. These findings are somewhat dissimilar to other country samples; for example, in the 21 countries that participated in the WMHS, physical abuse (29.3%), family violence (24.8%) and neglect (19.3%) were the most prevalent childhood adversities among those with a lifetime suicide attempt, while physical abuse (20.6%), family violence (17.6%) and death of a parent (14.2%) were most often reported among participants with lifetime suicidal ideation.[20] Cross-nationally, it would appear that physical abuse is the commonest childhood adversity associated with lifetime suicide attempts and ideation.[20]

The estimate lifetime prevalence of 2.9% for attempted suicide among South Africans is close to the rates of 4.6% and 4.1% reported for general and black populations, respectively, in the USA. In addition, the 9.1% estimated prevalence of suicide ideation is comparable with previous estimates from studies in South African clinical samples. Joe et al[6] reported for the first time on the rates of suicide ideation, plan and attempts among the different ethnic groups, in data from the SASH study. Overall, the results suggest that people in South Africa engage in suicidal thought and behaviours at levels nearly comparable with those of Western nations.

When examining suicidal behaviour, risk in the context of childhood adversity, sexual abuse, physical abuse and parental divorce emerged as significant risk factors for lifetime suicide attempts in the total sample. Furthermore, physical abuse and parental divorce were significant risk factors for suicidal ideation in the total sample. After adjusting for mental illness, sexual abuse was also a significant risk factor for suicidal ideation. Parental divorce emerged as a significant risk factor among ideators with lifetime suicide. These findings are largely consistent with the data from the overall cross-national WMHS, which found that physical and sexual abuse significantly increased the likelihood of suicidal ideation and attempts, while neglect was a risk factor for suicidal behaviour in multivariate additive analyses.[20]

**Table 5** Final multivariate model for associations between childhood adversities and lifetime (LT) suicidality*

| | LT attempts in total sample† | | Ideators among total sample† | | Ideators with LT plans‡ | | Ideators with LT attempts§ | |
|---|---|---|---|---|---|---|---|---|
| | OR (95% CI) | $\chi^2$ | OR (95% CI) | $\chi^2$ | OR (95% CI) | $\chi^2$ | OR (95% CI) | $\chi^2$ |
| Physical abuse | 2.2 (1.3 to 3.8)¶ | 8.9 (0.003)¶ | 2.1 (1.6 to 2.8)¶ | 25.4 (<0.001)¶ | 0.4 (0.2 to 1.0)¶ | 4.3 (0.038)¶ | 0.8 (0.3 to 2.1) | 0.3 (0.60) |
| Sexual abuse | 9.3 (2.5 to 35.2)¶ | 11.2 (<0.001)¶ | 3.7 (0.9 to 15.9) | 3.3 (0.07) | – | – | – | – |
| Parent died | 1.2 (0.7 to 2.3) | 0.4 (0.51) | 1.7 (1.1 to 2.6)¶ | 6.6 (0.010)¶ | 0.4 (0.1 to 1.3) | 2.2 (0.14) | 0.6 (0.3 to 1.1) | 2.8 (0.10) |
| Parent divorced | 3.1 (1.7 to 5.6)¶ | 14.5 (<0.001)¶ | 1.9 (1.2 to 3.0)¶ | 8.1 (0.004)¶ | 0.7 (0.2 to 2.3) | 0.4 (0.51) | 2.4 (0.9 to 6.4) | 3.0 (0.08) |
| Other parent loss | 1.1 (0.3 to 4.3) | 0.0 (0.87) | 2.1 (1.3 to 3.6)¶ | 8.3 (0.004)¶ | 0.3 (0.0 to 2.0) | 1.8 (0.18) | 1.3 (0.1 to 13.3) | 0.1 (0.79) |
| Family violence | 0.9 (0.3 to 2.3) | 0.1 (0.76) | 1.1 (0.6 to 2.3) | 0.2 (0.69) | 0.4 (0.1 to 1.8) | 1.6 (0.20) | 1.2 (0.4 to 4.1) | 0.1 (0.76) |
| Physical illness | 1.4 (0.4 to 5.3) | 0.2 (0.63) | 1.6 (0.7 to 3.3) | 1.4 (0.24) | 0.6 (0.1 to 2.5) | 0.5 (0.46) | 0.9 (0.2 to 3.3) | 0.0 (0.85) |
| Financial adversity | 1.3 (0.4 to 3.7) | 0.2 (0.65) | 0.9 (0.4 to 1.7) | 0.1 (0.71) | 1.6 (0.4 to 6.0) | 0.6 (0.44) | 1.4 (0.5 to 4.3) | 0.4 (0.52) |
| Group significance test for all types | | 29.4 (<0.001)¶ | | 43.0 (<0.001)¶ | | 833.9 (<0.001)¶ | | 11.5 (0.18) |
| Significance test for difference between types | | 13.1 (0.07) | | 9.2 (0.24) | | 805.7 (<0.001)¶ | | 11.8 (0.11) |
| 2+ adversities | 0.7 (0.2 to 1.8) | 0.7 (0.41) | 0.5 (0.3 to 0.9)¶ | 4.9 (0.028)¶ | 4.7 (0.8 to 29.2) | 2.9 (0.09) | 2.9 (0.8 to 10.6) | 2.7 (0.10) |
| Group significant test of controls: demographic variables | | 414.6 (<0.001)¶ | | 1112.0 (<0.001)¶ | | 214.0 (<0.001)¶ | | 10.4 (0.005)¶ |
| Group significance of controls: demographics and interactions between demographics and intervals | | 831.6 (<0.001)¶ | | 1405.7 (<0.001)¶ | | 1063.6 (<0.001)¶ | | 174.7 (<0.001)¶ |
| Group significance of controls: demographics and interactions between demographics and intervals | | 8596.4 (<0.001)¶ | | 6292.0 (<0.001)¶ | | 4268.0 (<0.001)¶ | | 532.5 (<0.001)¶ |
| Group significance of controls: parent psychopathology | | 11.6 (0.003)¶ | | 15.4 (<0.001)¶ | | 15.0 (0.020)¶ | | |

*Assessed in part 2 sample due to having part 2 controls. Controls for the model include intervals (1–5 intervals), and also include significant variables from demographic and parent psychopathology models, details in the following footnotes.
†Model controls for intervals (1–5 intervals), demographics (sex, age, time-varying education), interaction between intervals (13–19, 20–29, 30+) and age, education. For parent psychopathology, controlling for number of parental disorders (dummies for 1, 2+ disorders).
‡Model controls for intervals (1–5 intervals), demographics (sex, age, time-varying education), interaction between intervals (13–19, 20–29, 30+) and age, education. For parent psychopathology, controlling for types of parental disorders (6 dummies).
§Model controls for intervals (1–5 intervals), demographics (sex, age, time-varying education), interaction between intervals (13–19, 20–29, 30+) and age, education. Parent psychopathology not controlled for due to insignificance in previous models.
¶Significant at the 0.05 level, two-sided test.

**Table 6** Final multivariate model for associations between childhood adversities and lifetime (LT) suicidality, controlling for mental disorders*

| | LT attempts in total sample† | | Ideators among total sample† | | Ideators with LT plans‡ | | Ideators with LT attempts§ | |
|---|---|---|---|---|---|---|---|---|
| | OR (95% CI) | $\chi^2$ | OR (95% CI) | $\chi^2$ | OR (95% CI) | $\chi^2$ | OR (95% CI) | $\chi^2$ |
| Physical abuse | 2.1 (1.2 to 3.7)¶ | 6.1 (0.013)¶ | 2.0 (1.5 to 2.8)¶ | 19.7.4 (<0.001)¶ | 0.4 (0.2 to 0.9)¶ | 5.7 (0.017)¶ | 0.7 (0.3 to 1.9) | 0.4 (0.53) |
| Sexual abuse | 11.7 (3.3 to 42.3)¶ | 14.7 (<0.001)¶ | 4.6 (1.2 to 18.1)¶ | 4.9 (0.027)¶ | – | – | – | – |
| Parent died | 1.3 (0.7 to 2.4) | 0.6 (0.46) | 1.8 (1.2 to 2.7)¶ | 7.6 (0.006)¶ | 0.4 (0.1 to 1.3) | 2.5 (0.11) | 0.5 (0.3 to 1.1) | 3.3 (0.07) |
| Parent divorced | 3.4 (1.8 to 6.2)¶ | 15.6 (<0.001)¶ | 2.0 (1.2 to 3.1)¶ | 8.8 (0.003)¶ | 0.7 (0.2 to 2.5) | 0.2 (0.63) | 2.3 (0.8 to 6.5) | 2.7 (0.10) |
| Other parent loss | 1.2 (0.3 to 4.3) | 0.1 (0.79) | 2.1 (1.3 to 3.4)¶ | 8.9 (0.003)¶ | 0.3 (0.0 to 2.2) | 1.6 (0.20) | 1.5 (0.2 to 9.2) | 0.2 (0.66) |
| Family violence | 0.9 (0.3 to 2.7) | 0.0(0.89) | 1.1 (0.5 to 2.4) | 0.1 (0.75) | 0.5 (0.1 to 2.8) | 0.6 (0.42) | 1.4 (0.4 to 5.1) | 0.3 (0.59) |
| Physical illness | 1.4 (0.4 to 5.2) | 0.3 (0.61) | 1.4 (0.6 to 3.1) | 0.7 (0.41) | 0.6 (0.2 to 2.7) | 0.4 (0.54) | 1.0 (0.3 to 3.7) | 0.0 (0.98) |
| Financial adversity | 1.5 (0.5 to 4.5) | 0.6 (0.45) | 1.0 (0.5 to 2.0) | 0.0 (0.99) | 1.8 (0.5 to 6.7) | 0.8 (0.37) | 1.1 (0.3 to 3.8) | 0.0 (0.84) |
| Group significance test for all types | | 32.5 (<0.001)¶ | | 44.5 (<0.001)¶ | | 586.1 (<0.001)¶ | | 18.4 (0.018)¶ |
| Significance test for difference between types | | 16.2 (0.023)¶ | | 6.9 (0.44) | | 569.3 (<0.001)¶ | | 12.6 (0.08) |
| 2+ adversities | 0.7 (0.2 to 1.8) | 1.4 (0.25) | 0.4 (0.2 to 0.8)¶ | 6.4 (0.011)¶ | 3.2 (0.4 to 23.8) | 1.4 (0.24) | 2.6 (0.7 to 9.6) | 2.2 (0.14) |
| Group significant test of controls: demographic variables | | 408.3 (<0.001)¶ | | 1173.4 (<0.001)¶ | | 247.8 (<0.001)¶ | | 188.1 (<0.001)¶ |
| Group significance of controls: interactions between demographics and intervals | | 819.9 (<0.001)¶ | | 1303.7 (<0.001)¶ | | 1230.7 (<0.001)¶ | | 391.3 (<0.001)¶ |
| Group significance of controls: demographics and interactions between demographics and intervals | | 9644.8 (<0.001)¶ | | 5395.1 (<0.001)¶ | | 3699.2 (<0.001)¶ | | 1530.7 (<0.001)¶ |
| Group significance of controls: parent psychopathology | | 5.2 (0.07) | | 5.5 (0.06) | | 19.5 (0.003)¶ | | |
| Group significance of controls: mental disorders | | 121.9 (<0.001)¶ | | 131.1 (<0.001)¶ | | 33.5 (<0.001)¶ | | 29.5 (<0.001)¶ |

*Assessed in Part 2 sample due to having part 2 controls. Controls for the model include intervals (1–5 intervals), and also include significant variables from demographic and parent psychopathology models, details in the following footnotes.
†Model controls for intervals (1–5 intervals), demographics (sex, age, time-varying education), interaction between intervals (13–19, 20–29, 30+) and age, education. For parent psychopathology, controlling for number of parental disorders (dummies for 1, 2+ disorders).
‡Model controls for intervals (1–5 intervals), demographics (sex, age, time-varying education), interaction between intervals (13–19, 20–29, 30+) and age, education. For parent psychopathology, controlling for types of parental disorders (6 dummies).
§Model controls for intervals (1–5 intervals), demographics (sex, age, time-varying education), interaction between intervals (13–19, 20–29, 30+) and age, education. Parent psychopathology not controlled for due to insignificance in previous models.
¶Significant at the 0.05 level, two-sided test.

Of the adversities implicated, sexual and physical abuse were more significant risk factors than other adversities, highlighting the fact that intrusive and aggressive experiences in childhood may have more devastating and longer lasting effects.[58] This may be due to the extreme powerlessness and loss of control that such abuse causes, or to physically aggressive assaults resulting in the devaluation of one's body and consequent susceptibility to self-harm.[28] In a country with high rates of sexual and physical abuse,[46] this is a matter of particular concern. The impact of parental divorce on suicidality supports previous findings that parental divorce, if accompanied by other adversities such as childhood abuse, increases the risk of suicidal behaviour.[59]

We also found that exposure to *two or more childhood adversities* significantly increased the risk of suicide attempts among ideators. This confirms earlier work showing that exposure to multiple childhood adversities increases the risk of suicidal behaviour.[21 23 24 60 61] Bruffaerts *et al*[20] found a subadditive effect with regard to the onset of suicidal behaviour when considering multiple adversities. Thus, the impact of multiple adversities was not equal to the sum of the ORs of individual adversities. In the overall WMHS analysis, exposure to multiple childhood adversities had a significant effect on the persistence of suicide when considering exposure to every additional childhood adversity; however, in the current study, it was not possible to stratify the number of adversities beyond two or more (ie, into more than two categories), given the relatively small number of cases in the sample overall with non-fatal suicidal behaviour. Physical abuse, parental death, parental loss other than through death and parental divorce emerged as independent risk factors for suicidal ideation in the total sample. Moreover, the effects of childhood adversities on suicidal tendencies tended to differ over the *life course*. Consistent with nationally representative data in WMHS, childhood adversities were associated with the highest risk of suicide attempts in childhood, with a decrease in risk in adolescence and young adulthood, followed by an increase in risk again during later adulthood.[20]

In *childhood*, sexual abuse was significantly associated with lifetime suicide attempts in the total sample, while sexual and physical abuse were significantly associated with suicidal ideation. Among suicidal ideators, parental death was significantly associated with lifetime suicide attempts. Exposure to childhood sexual abuse, physical abuse or parental divorce significantly increased suicide attempts during *teen years*, while physical abuse and parental death were associated with suicidal ideation in teens. Among teen suicidal ideators, physical illness was significantly associated with suicidal plans, while parental divorce was associated with suicide attempts. These findings emphasise the need to focus particularly on suicide prevention strategies at youth . In *young adulthood*, parental loss other than the death of a parent was significantly associated with suicidal ideation in the total sample.

Interestingly, childhood physical abuse was identified as a significant risk factor for suicidal attempts in *later adulthood*, while childhood physical illness and parental loss other than the death of a parent significantly increased the risk for ideation.

Similar to findings from SASH, childhood sexual abuse emerged as a particularly robust risk factor for suicide attempts in younger participants in the WMHS cross-national analysis, with a 10.9 times higher OR of suicide attempts in children, a 6.1 times higher likelihood in adolescents and a 2.9-fold risk in young adults who were exposed.[20] This is in keeping with the Enns hypothesis that sexual abuse results in suicidal behaviour at a younger age.[21] Consistent with other studies, childhood physical and sexual abuse, in particular, emerged as risk factors for the emergence and persistence of suicidal behaviour, especially in adolescence. Loss of a parent, physical ill-health and family violence have also been found to be associated with persistence of suicidality.[20 28 58] These findings extend previous work carried out in other developing countries that have found childhood adversities to be a significant risk factor for suicidality.[20 62–64]

## Limitations

The following limitations need to be highlighted. First, recall bias might have impacted on the accuracy of recall of childhood adversities. That said, participants were asked questions about childhood adversities in sequence which may have facilitated more accurate recall.[65] Systematic reviews have also found that recall of past experiences can be accurate and provide valuable data.[66 67] Thus, there is evidence to support the validity of accurate recall of childhood adversities.[67] Furthermore, studies have shown that responses to questions on childhood adversities, similar to those asked in the SASH study, generally remain stable over time.[68 69] We recommend that future studies examine ethnicity in relation to adversity and suicidal outcomes. Second, in view of the cross-sectional design, more detailed, temporal information on childhood adversities and suicidal incidents was not obtained. Third, variables such as culture, ethnicity and mental status at the time of the interview may have influenced the recall and reporting of suicidal behaviour. It is possible that response bias may have been particularly skewed to disenfranchised South Africans (eg, poor, young, urban and black respondents), who may have been too afraid to divulge information on suicidality. Stigma associated with mental health problems may have also played a role in reporting suicidal tendencies. Thus, participants' mental health status, ethnicity, culture and generational factors may have also contributed to the under-reporting of suicidality. It is possible that individuals reporting childhood adversities may have also been more likely to report suicidal behaviour, while those not reporting childhood adversities may have under-reported suicidality. Stigma and mental health status (eg, depressed persons may be more inclined to report suicidality and

more likely to remember negative childhood experiences) may also be contributory factors. In addition, some participants may have been afraid to report suicidal behaviours. The role of ethnicity, culture and generational factors may have also contributed to the under-reporting of suicidality. Overall, it is much more likely that adversities and suicidality were under-reported rather than over-reported.[9 20 67 70] Fourth, we did not assess for self-mutilating behaviour. The importance of discriminating suicidal behaviour from non-suicidal self-mutilation cannot be underestimated. Fifth, the survey was conducted in adults living in households and hostel quarters; thus, the findings are not generalisable to homeless and institutionalised persons who were not included in the survey. Sixth, the CIDI instrument which was used in this study is a lay-administered instrument which does not include an assessment of several key Diagnostic and Statistical Manual of Mental Disorders (DSM)-IV diagnoses (such as bipolar disorder and psychosis), which are associated with elevated rates of suicidality. As a result, some participants with suicidality may not have been diagnosed as those with a disorder. Furthermore, in view of the large CIs and small sample sizes for some of these analyses, caution is required in drawing conclusions. In addition, we did not control for other unmeasured causes of childhood adversities and suicidality, or protective (resiliency) factors that may have contributed to the associations observed in these data. Other risk and resiliency factors may have contributed to the prevalence of non-fatal suicidal behaviours and to the associations with different forms of childhood adversity and warrant further investigation. Finally, it is important to point out that these data were collected approximately 10 years ago. Notwithstanding these limitations, this study represents the first investigation among South Africans of a wide range of childhood adversities and their impact on the onset and persistence of suicidality over the life course.

## Conclusions

Childhood adversities, especially sexual abuse, physical abuse and parental divorce, are associated with the onset and persistence of suicidal behaviour with the risk greatest in children and adolescents. Public health efforts aimed at prevention of early childhood sexual and physical abuse, in particular, may have a significant impact on reducing suicidality over the life course and improving mental health outcomes.

**Author affiliations**
[1]Department of Psychiatry, Stellenbosch University, Tygerberg, Republic of South Africa
[2]Department of Society, Human Development and Health, Harvard School of Public Health, Harvard University, Boston, Massachusetts, USA
[3]Department of African and African American Studies, Harvard University, Boston, Massachusetts, USA
[4]Department of Psychiatry and Mental Health, University of Cape Town, Cape Town, Republic of South Africa

**Contributors** BB and RG were involved in data interpretation, drafting manuscript, final approval of manuscript submitted for publication, ensuring that questions related to the accuracy of the work are appropriately resolved. MB was involved in data interpretation, revising the manuscript, final approval of manuscript submitted for publication, ensuring that questions related to the accuracy of the work are appropriately resolved. DRW, DJS and SS provided substantial contributions to the conception or design of the work, data acquisition, data analysis and interpretation, critically revising the manuscript, final approval of the version to be published and accountability for all aspects of the work.

**Funding** This work was funded by the South African Department of Health and the University of Michigan, Medical Research Council of South Africa. This study was also funded by NIH. Additional funding was received from the South African Department of Health and the University of Michigan.

**Competing interests** DJS and SS are supported by the Medical Research Council of South Africa. DJS has received research grants and/or consultancy honoraria from AstraZeneca, Eli Lilly, GlaxoSmithKline, Lundbeck, Orion, Pfizer, Pharmacia, Roche, Servier, Solvay, Sumitomo and Wyeth. SS has received research grants or travel sponsorship from AstraZeneca, Eli Lilly, GlaxoSmithKline, Lundbeck and Servier. SS is supported by the South African Research Chairs Initiative of the Department of Science and Technology and the National Research Foundation. BB has received congress sponsorship from Janssen-Cilag.

**Ethics approval** Human Subjects Committee of the University of Michigan, Harvard Medical School ethics committee and a single project assurance of compliance from the Medical University of South Africa (MEDUNSA), and the National Institute of Mental Health.

**Provenance and peer review** Not commissioned; externally peer reviewed.

**Data sharing statement** No additional data are available.

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
