## [Reviewer comments · BMJ Open]

Some articles will have been accepted based in part or entirely on reviews undertaken for other BMJ Group journals. These will be reproduced where possible.

ARTICLE DETAILS

TITLE (PROVISIONAL)	Association between childhood adversities and long-term suicidality among South Africans from the results of the South African Stress and Health Study: a cross-sectional study
AUTHORS	Bruwer, Belinda; Govender, Ravi; Bishop, Melanie; Williams, David; Stein, Dan; Seedat, Soraya

VERSION 1 - REVIEW

REVIEWER	Gerard Leavey University of ULster Northern Ireland
REVIEW RETURNED	08-Jan-2014

GENERAL COMMENTS	For an international audience, perhaps not familiar with apartheid classifications, the authors could define the racial categories used, particularly "coloured". They also need to comment as to the proportions of these groups in the general population in SA. On this point also, I am surprised at the failure to examine ethnicity in relation to adversity and the suicide outcome variables. was this considered too sensitive? How was urbanity and rurality defined? And, did the variable relate to childhood or adulthood residency? The authors should say more about potential response bias - the likelihood of non-response among young, urban, poor and black. The statistical reporting in the findings should be consistent - sometimes confidence intervals for OR are given and other times, not. Also, could you check the OR for Physical abuse, given as 0.4? Overall, the paper could do with considerable reduction in reporting the findings.
--

REVIEWER	Ingunn Rangul Askeland Norwegian Centre for Violence and Traumatic Stress studies- NKVTS Norway
REVIEW RETURNED	13-Jan-2014

GENERAL COMMENTS	Thank you for an important contribution on how childhood adversities have an severe impact on peoples life and presenting such a large amount of data from a country outside the US and the Europeand contries.
---

The authors have examined associations between eight different childhood adversities and long-term suicidal behaviour among South African adults. The study aimed to examine the relationship between both type and frequency of childhood adversities and suicidal behaviour. A probability sample of 4,351 participants was interviewed on childhood adversities, diagnostic information, suicidal ideation, suicidal plans and suicide attempts. The results showed a high prevalence of both childhood adversities and suicidal behaviour in this sample. Physical abuse, sexual abuse and parental divorce were significantly associated with lifetime suicide attempts. Physical abuse and parental divorce were associated with suicide ideation, and two or more childhood adversities were associated with a 2-fold higher risk of lifetime suicide attempts.

The authors address an important topic. The article is generally well written and offers an extensive and thorough amount of data. However, some clarifications might improve the manuscript further. In the abstract section (p. 2, line 24) the authors describe that the respondents provide diagnostic information. This is mentioned again under strengths and limitations as “mental status” p. 4, line 19 and 34), and under the description of CIDI (line 48-58). Psychiatric disorder as a risk factor for suicidality is further mentioned in the Introduction section (p. 6, line 26), in more detail on p. 9, line 16-25, and in the Data analysis section (p.11, line 30, 32). The focus in this article is on childhood adversities and their associations to suicidal behaviour. However, psychiatric disorders are both mentioned in the discussion section and data on psychiatric disorders are measured. The authors points to an interesting discussion on what seems to be important factors in the aetiology of suicidal behaviour. Thus, the article might benefit from some more elaboration on how information

on mental health was used in the analysis. Further, it is somewhat unclear what constituted the differences between respondent's mental status and parental psychopathology, regarding measurement and analysis. The authors might consider elaborating this topic.

In the Strengths and limitation section the authors reflects on how mental health might have contributed to the reporting of suicidality (e.g. p. 4., line 34). It might also be that the people reporting childhood adversities also tend to report suicidality and those not reporting childhood adversities tend to underreport suicidality. Another factor could be related to their mental health status, e.g. depressed persons may be more prone to report suicidality and more likely to remember negative childhood experiences. This is possible factors related to reporting that the authors might want to include several places in the article, both in regard to reporting suicidality and childhood adversities.

At page 8 line 25 (and p.8, line 49) the authors write "All racial and ethnic groups...". It is not clear what racial and ethnic groups the authors refers to since they are not previously described.

At p. 9, line 25-32, there is a description of how the percentages were weighted, this section might be more appropriate in the Data analysis section.

The description on how suicidal behaviour was measured is somewhat unclear. The authors might add some information on whether all the questions are taken from CIDI. I would also like a little more information on what questions from the CTS (p. 10, line 18) was used and which version of the CTS was administered. It is also not clear what the authors mean by the term "family violence",

	e.g. does this contain physical violence between parents or other acts of violent behaviour. In the Data analysis section (p.11, line 13) it might have been advisable to present an argument for choosing life stages and not keeping the age as a continuous variable. Additionally, adding some information on why these stages were chosen. In the Result section (p. 11, line 42-49) it would have been interesting to know how this sample is compared to the population in South Africa, e.g. in regard to percentages of black, coloured and so on. At the end, in the Conclusion section, the authors might consider phrasing the first sentence (p.22, line 33) a bit different and use the word associated with instead of “important risk factors”. The article contains a substantial amount of information and quite many tables. It might be an idea to reduce the numbers of tables and thereby make the message of the article more “sharpened” and accessible.
--	---

REVIEWER	Zainab Samaan McMaster University, Canada
REVIEW RETURNED	16-Jan-2014

GENERAL COMMENTS	General comments Although the manuscript is well written with pertinent background presented, there are many concerns including:  1. No psychiatric diagnoses provided and adjusted for in the analysis. Adverse events are associated with such disorders, suicidal behavior [SB] is commonly associated with psychopathology, it is therefore difficult to state with any certainty that adverse events and not psychiatric disorders are associated with SB. 2. The subgroups of SB into suicide attempts, ideation, and
---

ideation with and without plans are not justified, no rationale was provided for such groups and the relevance to the overall study objectives

3. The data are 10 years old and this is a limitation needed to be included in the discussion
4. The exact sample size should be provided for each subgroup and variable. The reporting of percentages is misleading as this assumes complete data for every variable
5. The authors used four or five outcomes, yet no adjustment for multiple testing error.
6. Missing data handling should also be reported, this is related to point 4 above.
7. Since the data are already collected, the authors should report an estimate of power for the given sample size to test primary hypothesis
8. A discussion about the difference between suicide attempts and self harm should also be considered since there was no question about intent to die in the questions posed
9. Why using bivariate model and multivariate analyses? Bivariate did not add any relevant results but merely a repetition.
10. The reporting of percentage of adverse events in the various subgroups of SB is confusing. For example page 10 of the results stated: 35% of those with one adversity made a suicide attempt compared with 23% with one adversity that did not make a suicide attempt. If 35% of the group with one adversity made a suicide attempt, the rest of this group [65%] did not make an attempt? The same reporting is consistent throughout and should be revised.
11. Participants' flow diagram should be provided.

Specific comments

Abstract

Authors mentioned psychiatric diagnostic interviews, however no results were presented.

Introduction

Page 5, suicide risk in children 4-12 years of age, should this be framed to self harm? Do children as young as 4 have the ability to consider intent to die?

Methods

Page 8 suicidal behavior: the several subgroups are unclear and can not distinguish individuals with self harm but no intent to die

Page 9 childhood adversities: provide a reference to the CONFLICT scale and state in what way was it modified from its original form.

Gender: please replace with Sex. Gender is a social construct while sex is a biological construct, unless the authors assessed gender, I

	am assuming they mean the sex of the individuals. Acknowledgment Page 22 "DJS received research grants and/or consultancy", please be more specific. There is also a typo in this paragraph "SS IS SUPPORTED BY THE BY THE" Tables Table 1: provide the n for each variable Explain "matric" and "rands", income level categories and currency. Why do these household individuals have very high unemployment rate of 69%? Table 2 provide the total sample size for each subgroup and each cell in the table. Table 3, what is the superscript number 1 refers to in the title? Same for tables 4, 5 and 6. Why table 3 is needed?
--	--

VERSION 1 – AUTHOR RESPONSE

Reviewer Name Gerard Leavey

Institution and Country University of Ulster Northern Ireland

Please state any competing interests or state 'None declared': None declared

For an international audience, perhaps not familiar with apartheid classifications, the authors could define the racial categories used, particularly "coloured". They also need to comment as to the proportions of these groups in the general population in SA. On this point also, I am surprised at the failure to examine ethnicity in relation to adversity and the suicide outcome variables. was this considered too sensitive?

We would like to thank the reviewer for pointing this out. Race classification in South Africa has socio-political foundations. South Africa categorizes four race classes: Asian/Indian, Black Africans, Coloured and White. In South Africa, coloured refers to any person of "mixed-blood". We have clarified this in the manuscript [page 6, underneath the section "methods" and subsection "sampling approach" (para 2)]. According to 2001 Census statistics, 79% people in South Africa were Black African, 8.9% were coloured, 9.6% were white, and 2.5% were Indian/Asian (Statistics South Africa, 2001)

Statistics South Africa. (2001). Census 2001: Census in Brief. Pretoria: Statistics South Africa. Available from <http://www.statssa.gov.za/census01/html/CInBrief/CIB2001.pdf> (Accessed January 2014)

We did not include ethnicity in relation to adversity and suicide outcome variables in the present study. The aim of this study was to examine the relationship between the type and frequency of childhood adversity exposure to suicidal behavior at different stages of the life course; as such, we did not examine the socio-demographic variables, including race, as correlates or predictors. We have in previous publications from this study reported on racial differences in the rates of attempted suicide (for example, Coloureds [mixed racial origin] endorsed levels of attempted suicide that were approximately three times higher than any other race groups (Joe et al., 2008)). We have also

reported differences in the experiences of childhood adversity by race and first-onset and lifetime mental disorders (Slopen et al., 2010, Seedat et al., 2009).

Joe S, Stein DJ, Seedat S, Herman A, Williams DR. (2008) Non-fatal suicidal behavior among South Africans :results from the South Africa Stress and Health Study. Soc Psychiatry Psychiatr Epidemiol, 43(6):454-61.

Slopen N, Williams DR, Seedat S, Moomal H, Herman A, Stein DJ. (2010) Adversities in childhood and adult psychopathology in the South Africa Stress and Health Study: associations with first-onset DSM-IV disorders. Soc Sci Med, 71(10),1847-54.

Seedat S, Stein DJ, Jackson PB, Heeringa SG, Williams DR, Myer L. (2009) Life stress and mental disorders in the South African stress and health study. S Afr Med J 99(5), 375-82.

Although, we agree that this would have been an interesting aspect to consider in future, the study was not adequately powered to assess ethnicity as a predictor of suicidality by life stage.

We have included as a limitation the following: "Variables such as culture, ethnicity and mental status at the time of the interview may have influenced the recall and reporting of suicidal behaviour" We have also highlighted this section within the manuscript for the reviewers (see page 21, underneath subsection "limitations").

How was urbanity and rurality defined? And, did the variable relate to childhood or adulthood residency?

Urbanity and rurality was defined according to the 2001 South African Census enumeration areas (South African Census, 2001). The reference to urbanity and rurality refers to residency of the adults in the sample (i.e. the sample was collected from adults living in urban and rural settings).

Classification type according to the Census 2001 EA

EA Type Geography Type Urban/Rural

Vacant

Small Holding

Urban Settlement

Recreational

Industrial Area

Institution

Hostel

Urban-Formal

Urban

Informal Settlement Urban-Informal

Farm

Small Holding

Recreational

Industrial Area

Institution

Hostel

Rural-Formal

Rural

Vacant

Tribal Settlement

Recreational

Industrial Area
Hostel

Tribal-Area

South African Census (2001) Investigation into appropriate definitions of urban and rural areas for South Africa: Discussion document/ Statistics South Africa. [Report No. 03-02-20 (2001)]. Pretoria: Statistics South Africa, 2003, 195p. Available from:

<http://www.statssa.gov.za/census01/html/UrbanRural.pdf> (Accessed January 2014)

Please also see the following article for a description of the SASH rationale and design:

Williams DR, Herman A, Kessler RC, Sonnega J, Seedat S, Stein DJ, Moomal H, Wilson CM. (2004). The South Africa stress and health study: rationale and design. *Metabolic Brain Disease*, 19, 135–147. [PubMed: 15214513]

A South African sample of 4,351 adults who live in hostel quarters and households were included in the study. The sampled residences were stratified into ten housing categories: rural-commercial, agricultural, rural traditional subsistence areas, African townships, informal urban or peri-urban shack areas, Coloured townships, Indian townships, general metropolitan residential areas, general large metropolitan residential areas, and domestic servant. We have included the above in the methods section of the manuscript (under 'sampling approach').

The authors should say more about potential response bias - the likelihood of non-response among young, urban, poor and black.

We included potential response bias as the following limitations in the manuscript:

Variables such as culture, ethnicity and mental status at the time of the interview may have influenced the recall and reporting of suicidal behavior. It is possible that response bias may have been particularly skewed to disenfranchised South Africans (e.g. poor, young, urban and black respondents), who may have been too afraid to divulge information on suicidality. Stigma associated with mental health problems may have also played a role in the reporting suicidal tendencies. Thus, participants' mental health status, ethnicity, culture and generational factors may have also contributed to the under-reporting of suicidality (please see page 21 (paragraph 2) and p 22 (top section of page), underneath subsection "Limitation").

The statistical reporting in the findings should be consistent - sometimes confidence intervals for OR are given and other times, not. Also, could you check the OR for Physical abuse, given as 0.4?

We have included missing CIs in the text. The OR for physical abuse was 0.4. We have clarified this in the manuscript.

Overall, the paper could do with considerable reduction in reporting the findings.

Thank you for pointing this out. We have rationalized this and combined tables to make the findings more digestible and reader friendly.

Reviewer Name Ingunn Rangul Askeland

Institution and Country Norwegian Centre for Violence and Traumatic Stress studies- NKVTS Norway

Please state any competing interests or state 'None declared': None declared

If you have any further comments for the authors please enter them below.

Thank you for an important contribution on how childhood adversities have an severe impact on peoples life and presenting such a large amount of data from a country outside the US and the Europeand contries.

The authors have examined associations between eight different childhood adversities and long-term suicidal behaviour among South African adults. The study aimed to examine the relationship between both type and frequency of childhood adversities and suicidal behaviour. A probability sample of 4,351 participants was interviewed on childhood adversities, diagnostic information, suicidal ideation, suicidal plans and suicide attempts. The results showed a high prevalence of both childhood adversities and suicidal behaviour in this sample. Physical abuse, sexual abuse and parental divorce were significantly associated with lifetime suicide attempts. Physical abuse and parental divorce were

associated with suicide ideation, and two or more childhood adversities were associated with a 2-fold higher risk of lifetime suicide attempts.

The authors address an important topic. The article is generally well written and offers an extensive and thorough amount of data. However, some clarifications might improve the manuscript further. In the abstract section (p. 2, line 24) the authors describe that the respondents provide diagnostic information. This is mentioned again under strengths and limitations as "mental status" p. 4, line 19 and 34), and under the description of CIDI (line 48- 58). Psychiatric disorder as a risk factor for suicidality is further mentioned in the Introduction section (p. 6, line 26), in more detail on p. 9, line 16-25, and in the Data analysis section (p.11, line 30, 32). The focus in this article is on childhood adversities and their associations to suicidal behaviour. However, psychiatric disorders are both mentioned in the discussion section and data on psychiatric disorders are measured. The authors points to an interesting discussion on what seems to be important factors in the aetiology of suicidal behaviour. Thus, the article might benefit from some more elaboration on how information on mental health was used in the analysis.

The primary aim of this study was to examine the relationship between the type and frequency of childhood adversity exposure to suicidal behavior at different stages of the life course. We have in previous publications reported on the prevalence and correlates of suicidal behavior in relation to mental disorders (Joe et al., 2008) as well as on the differences in the experiences of childhood adversity by race and first-onset and lifetime mental disorders (Slopen et al., 2010, Seedat et al., 2009).

Joe S, Stein DJ, Seedat S, Herman A, Williams DR. (2008) Non-fatal suicidal behavior among South Africans :results from the South Africa Stress and Health Study. *Soc Psychiatry Psychiatr Epidemiol*, 43(6):454-61.

Slopen N, Williams DR, Seedat S, Moomal H, Herman A, Stein DJ. (2010) Adversities in childhood and adult psychopathology in the South Africa Stress and Health Study: associations with first-onset DSM-IV disorders. *Soc Sci Med*, 71(10),1847-54.

Seedat S, Stein DJ, Jackson PB, Heeringa SG, Williams DR, Myer L. (2009) Life stress and mental disorders in the South African stress and health study. *S Afr Med J* 99(5), 375-82.

Further, it is somewhat unclear what constituted the differences between respondent's mental status and parental psychopathology, regarding measurement and analysis. The authors might consider elaborating this topic.

We refer to our response above. The relationship between parental psychopathology and mental disorders in the offspring (i.e the respondents in this study have been reported in a paper by McLaughlin et al (2012) which reports collectively on data from the World Mental Health Surveys, including South Africa.

McLaughlin KA, Gadermann AM, Hwang I, Sampson NA, Al-Hamzawi A, Andrade LH, Angermeyer MC, Benjet C, Bromet EJ, Bruffaerts R, Caldas-de-Almeida JM, de Girolamo G, de Graaf R, Florescu S, Gureje O, Haro JM, Hinkov HR, Horiguchi I, Hu C, Karam AN, Kovess-Masfety V, Lee S, Murphy SD, Nizamie SH, Posada-Villa J, Williams DR, Kessler RC. (2012). Parent psychopathology and offspring mental disorders: results from the WHO World Mental Health Surveys. *Br J Psychiatry*, 200(4), 290-9.

In the Strengths and limitation section the authors reflects on how mental health might have contributed to the reporting of suicidality (e.g. p. 4., line 34). It might also be that the people reporting childhood adversities also tend to report suicidality and those not reporting childhood adversities tend to underreport suicidality. Another factor could be related to their mental health status, e.g. depressed persons may be more prone to report suicidality and more likely to remember negative childhood experiences. This is possible factors related to reporting that the authors might want to include several places in the article, both in regard to reporting suicidality and childhood adversities.

We would like to thank the reviewer for pointing this out. We have noted this and included it under the limitations (please see page 22).

At page 8 line 25 (and p.8, line 49) the authors write "All racial and ethnic groups...". It is not clear what racial and ethnic groups the authors refer to since they are not previously described.

Please see response to reviewer Gerard Leavey's first comment in the above section

At p. 9, line 25-32, there is a description of how the percentages were weighted, this section might be more appropriate in the Data analysis section.

We moved this section from 'Diagnostic Interview' to 'Data analysis'

The description on how suicidal behaviour was measured is somewhat unclear. The authors might add some information on whether all the questions are taken from CIDI.

Under "methods", there is a section on 'suicidal behavior' that describes how suicidal behavior was measured. We have only described the CIDI (which was a modified version of the CIDI), as this was the measurement instrument used to measure suicidal behaviour. Examples of questions are also given. We have now included more detail in this section (please see page 8 and 9).

I would also like a little more information on what questions from the CTS (p. 10, line 18) was used and which version of the CTS was administered.

A revised version of the CTS, namely the CTS2, was used (Straus, 1979; Straus, 2004; Straus et al., 1996).

Straus, M.A. (2004). Cross-cultural reliability and validity of the Revised Conflict Tactics Scales: a study of university student dating couples in 17 nations. *Cross-Cultural Res*; 38: 407–432.

Straus MA, Hamby SL, Boney-McKoy S, Sugarman DB (1996). The Revised Conflict Tactics Scales (CTS2): Development and Preliminary Psychometric Data. *Journal of Family Issues*, 17: 283–316.

Straus MA. Measuring Intrafamily Conflict and Violence: The Conflict Tactics (CT) Scales. (1979) *Journal of Marriage and Family*, 41(1):75

It is also not clear what the authors mean by the term "family violence", e.g. does this contain physical violence between parents or other acts of violent behaviour.

For the assessment of sexual violence, the following questions were asked: "The next 2 questions are about sexual assault: (i) The first is about rape. We define this as someone either having sexual intercourse with you or penetrating your body with a finger or object when you did not want them to, either by threatening you or using force, or when you were so young that you didn't know what was happening. Did this ever happen to you?", and (ii) "Other than rape, were you ever sexually assaulted or molested?". A modified version of the Conflict Tactics Scale (CTS2) was used to assess family violence and physical abuse (Straus, 1979). Respondents were classified as having experienced physical abuse when they indicated that, when they were growing up, their father or mother (includes biological, step, or adoptive parents) slapped, hit, pushed, grabbed, shoved, or threw something at them, or that they were beaten as a child by the persons who raised them. Family violence was assessed as present when respondents indicated that they (i) "were often hit, shoved, pushed, grabbed, or slapped while growing up" or (ii) "witnessed physical fights at home, like when your father beat up your mother?"

In the Data analysis section (p.11, line 13) it might have been advisable to present an argument for choosing life stages and not keeping the age as a continuous variable. Additionally, adding some information on why these stages were chosen.

Using this approach allowed us to examine the interactions between the life stage (13-19 years, 20-29 years, 30+ years) of respondents and each childhood adversity, as well as the influence each adversity had on early-, middle- and later- onset suicidality.

In the Result section (p. 11, line 42-49) it would have been interesting to know how this sample is compared to the population in South Africa, e.g. in regard to percentages of black, coloured and soon. Please see our response to a comment from reviewer Gerard Leavey on examining ethnicity in relation to adversity and suicide outcome variables.

At the end, in the Conclusion section, the authors might consider phrasing the first sentence (p.22, line 33) a bit different and use the word associated with instead of "important risk factors".

We have changed the reference to "risk factors" in the conclusion section of the manuscript given that this is a cross-sectional study and causality cannot be inferred.

The article contains a substantial amount of information and quite many tables. It might be an idea to reduce the numbers of tables and thereby make the message of the article more "sharpened" and

accessible.

We have tried our best to reduce the tables in order to make the representation of the results more digestible and reader friendly.

Reviewer Name Zainab Samaan

Institution and Country McMaster University, Canada Please state any competing interests or state 'None declared': None declared

General comments

Although the manuscript is well written with pertinent background presented, there are many concerns including:

1. No psychiatric diagnoses provided and adjusted for in the analysis. Adverse events are associated with such disorders, suicidal behavior [SB] is commonly associated with psychopathology, it is therefore difficult to state with any certainty that adverse events and not psychiatric disorders are associated with SB.

In mentioning the limitations, we now acknowledge that variables such as mental illness may have influenced the reporting of suicidal behaviour.

We have in previous publications reported on the prevalence and correlates of suicidal behavior in relation to mental disorders (Joe et al., 2008) as well as on the differences in the experiences of childhood adversity by race and first-onset and lifetime mental disorders (Slopen et al., 2010, Seedat et al., 2009).

We did not control for other unmeasured causes of childhood adversities and suicidality, or protective (resiliency) factors that may have contributed to the associations observed in these data. Both other risk and resiliency factors may have contributed to both the prevalence of non-fatal suicidal behaviours and to the associations with different forms of childhood adversity and warrant further investigation

Joe S, Stein DJ, Seedat S, Herman A, Williams DR. (2008) Non-fatal suicidal behavior among South Africans :results from the South Africa Stress and Health Study. *Soc Psychiatry Psychiatr Epidemiol*, 43(6):454-61.

Slopen N, Williams DR, Seedat S, Moomal H, Herman A, Stein DJ. (2010) Adversities in childhood and adult psychopathology in the South Africa Stress and Health Study: associations with first-onset DSM-IV disorders. *Soc Sci Med*, 71(10),1847-54.

Seedat S, Stein DJ, Jackson PB, Heeringa SG, Williams DR, Myer L. (2009) Life stress and mental disorders in the South African stress and health study. *S Afr Med J* 99(5), 375-82.

2. The subgroups of SB into suicide attempts, ideation, and ideation with and without plans are not justified, no rationale was provided for such groups and the relevance to the overall study objectives We have included a description of this, because previous reviewers wanted more clarity on how the groups were delineated.

3. The data are 10 years old and this is a limitation needed to be included in the discussion

We included this as a limitation in the discussion section (please see page 23).

4. The exact sample size should be provided for each subgroup and variable. The reporting of percentages is misleading as this assumes complete data for every variable

Please see footnote b in Table 2: the % represents the percentage of people with the adversity among the cases with the outcome variable indicated in the column header. For example: the first cell is the % of those with physical abuse among those with attempts. In other words, among those with a suicidal attempt, 35% of them experienced one adversity. Among those who made no suicidal attempt, 23.4% experienced one adversity. Please also see the flow diagram for clarification.

5. The authors used four or five outcomes, yet no adjustment for multiple testing error.

We believe that we have applied the most appropriate statistical methods for addressing the objectives of this paper. All data analyses were processed and analysed centrally by a team of statisticians at the Harvard School of Public Health (Boston, USA)

6. Missing data handling should also be reported, this is related to point 4 above.

Missing data was imputed. All analyses accounted for the complex survey design using person-level weights that incorporated sample selection, non-response and post-stratification factors.

7. Since the data are already collected, the authors should report an estimate of power for the given sample size to test primary hypothesis

The sample size was statistically adequate to address the primary hypothesis. SASH sampling was a stratified first-stage sample of South Africa's 2001 Census enumeration areas (EAs) followed by a second-stage sample of dwelling units from each sample EA and finally a third-stage random selection of a single adult respondent in each selected sample dwelling unit. The SASH primary-stage sampling units (PSUs) were EA units defined for the 2001 Census of South Africa. South Africa's land area at the time was divided into 85,783 geographic EA units. Prior to the first stage of sample selection, each of these EAs was assigned to one of 53 strata based on the province in which it was located, its urban/rural status and the majority population group in the EA (African, Coloured, Indian, and white). A total primary stage sample size of 960 EA units was allocated to the 53 strata approximately in proportion to the total number of Census EAs in the stratum. Within each stratum, the allocated sample of EAs was selected with probability proportionate to the total 2001 Census count of adult population for the EA. SASH interviewers contacted each household in the sample of dwelling units and selected a single adult respondent at random using the Kish procedure for objective respondent selection. If the household or the selected respondent refused to be interviewed for SASH, a random replacement was drawn from the enumerative listing for the EA. This procedure of randomly replacing nonrespondent sample units implicitly introduces an adjustment for nonresponse. A total sample of 5089 households was selected. The weighted response rate (by sample size) was 85.5% of the designated respondents selected at random from the eligible persons in each sample household.

Sample sizes and final dispositions for the SASH study

Sampled households 5089

Designated adult responders 5089

Initial interviews from field 4434

Interview cooperation rate 87.1%

Final SASH cases for analysis 4351

Complete data interview rate 98.1%

8. A discussion about the difference between suicide attempts and self harm should also be considered since there was no question about intent to die in the questions posed?

We did not assess other self-harm/multilating behavior in the SASH study. This is now mentioned as a limitation and the importance of discriminating suicidality and other self-multilating behavior is also mentioned (please see bottom of page 22 and first sentence on page 23).

9. Why using bivariate model and multivariate analyses? Bivariate did not add any relevant results but merely a repetition.

We believe that we have applied the most appropriate statistical methods for addressing the objectives of this paper. The association between suicidality and childhood adversity was examined using discrete-time survival models with the analysis unit being person-years. Bivariate analyses (considering one adversity at a time) and multivariate analyses (considering all adversities simultaneously) were conducted. Two types of multivariate models were tested: multivariate additive models (simultaneously considering all childhood adversities) and multivariate interactive models (with number and type of childhood adversities experienced by each respondent included as dummy variables)"

Furthermore, we combined the tables showing the results of bivariate and multivariate analyses on the associations between childhood adversities and lifetime suicidality (previously tables 3 and 7, now only table 3). We also combined the reporting of these results and show this in the manuscript with tracked changes. With regards to childhood adversities and lifetime suicidality, multivariate analyses revealed an additional association with suicidal ideation, namely parental divorce (OR = 1.6, p=0.038).

10. The reporting of percentage of adverse events in the various subgroups of SB is confusing. For example page 10 of the results stated: 35% of those with one adversity made a suicide attempt compared with 23% with one adversity that did not make a suicide attempt. If 35% of the group with one adversity made a suicide attempt, the rest of this group [65%] did not make an attempt? The same reporting is consistent throughout and should be revised.

Please see footnote b in Table 2: the % represents the percentage of people with the adversity among the cases with the outcome variable indicated in the column header. For example: the first cell is the % of those with physical abuse among those with attempts. In other words, among those with a suicidal attempt, 35% of them experienced one adversity. Among those who made no suicidal attempt, 23.4% experienced one adversity. Please also see the flow diagram for clarification.

11. Participants' flow diagram should be provided.

We have added a flow diagram

Specific comments

Abstract

Authors mentioned psychiatric diagnostic interviews, however no results were presented.

The parent study assessed for psychiatric disorders, however the present manuscript focuses specifically on suicidality and not on other psychopathology. Please refer to the following article for results on common mental health disorders in South Africans (results from SASH):

Herman, A., Stein, D., Seedat, S., Heeringa, S., Moomal, H., & Williams, D. (2009). The South African Stress and Health (SASH) study: 12-month and lifetime prevalence of common mental disorders. *South African Medical Journal = Suid-Afrikaanse Tydskrif Vir Geneeskunde*, 99(5 Pt 2), 339-344.

Please also see the following article on results from SASH with regards to childhood adversities and adult psychopathology:

Slopen, N., Williams, D., Seedat, S., Moomal, H., Herman, A., & Stein, D. (2010). Adversities in childhood and adult psychopathology in the South Africa Stress and Health Study: associations with first-onset DSM-IV disorders. *Social Science & Medicine* (1982), 71(10), 1847-1854.

doi:10.1016/j.socscimed.2010.08.015

Introduction

Page 5, suicide risk in children 4-12 years of age, should this be framed to self harm? Do children as young as 4 have the ability to consider intent to die?

We did not specifically assess for self harm. The study asked participants about suicidal behavior before age 12, and not self harm. According to the participants in the study, they experienced suicidal behavior before age 12. They were not asked if they had specifically experienced suicidal behaviour at age 4, but between age 4 and 12. We cannot, from the results of this study, say that a person at specifically age 4 considered intent to die, but we can say that they experienced suicidal behavior between age 4 and 12. "The intent of this study was specifically to examine the prevalence and associations of retrospectively reported childhood adversities, by type, with suicidal behaviours over the life course". We have acknowledged the possibility of recall bias of childhood adversities under the "limitations".

Methods

Page 8 suicidal behavior: the several subgroups are unclear and can not distinguish individuals with self harm but no intent to die Page 9 childhood adversities: provide a reference to the CONFLICT scale and state in what way was it modified from its original form.

We have now clarified the items/questions relating to the Conflict Tactics Scale (Straus, 1979) that were used to assess exposure to family violence (please see bottom of page 8, and first paragraph on page 9).

Straus MA. Measuring Intrafamily Conflict and Violence: The Conflict Tactics (CT) Scales. (1979) *Journal of Marriage and Family*, 41(1):75

Gender: please replace with Sex. Gender is a social construct while sex is a biological construct, unless the authors assessed gender, I am assuming they mean the sex of the individuals.

This has been changed throughout the manuscript.

Acknowledgment Page 22 "DJS received research grants and/or consultancy", please be more

specific. There is also a typo in this paragraph “SS IS SUPPORTED BY THE BY THE”
 This has been clarified and the typo corrected.

Tables

Table 1: provide the n for each variable.

We have included the n for each variable in this table and indicated these changes in red font (please see table 1).

Explain “matric” and “rands”, income level categories and currency.

Matric refers to people in Grade 12. Rands refer to the South African currency. Income level categories refer to the annual income of a household (in Rand).

Why do these household individuals have very high unemployment rate of 69%?

We agree that this is quite a high unemployment rate. This might be explained by the fact that most of the sample lives in an Urban area (59.7%0, and most had less than 12 years of education (62.7%).

There were slightly more females than males, and according to the 2011 SA Census, the unemployment rates are higher in females than males, namely 46% versus 34.6%, respectively. In South Africa, the rate of unemployment is also the highest amongst Black African women, namely 52.9% (76% of the sample were Black Africans).

Table 2 provide the total sample size for each subgroup and each cell in the table.

The data is person-year-level, therefore the Ns are represented in the row indicated by superscript a

Table 3, what is the superscript number 1 refers to in the title? Same for tables 4, 5 and 6.

This is indicated underneath the tables as footnotes

Why table 3 is needed?

Table 3 gives the results of the multivariate, and table 7 of the bivariate. We have combined these tables

VERSION 2 – REVIEW

REVIEWER	Ingunn Rangul Askeland Norwegian centre for violence and traumatic stress studies Norway
REVIEW RETURNED	18-Mar-2014

GENERAL COMMENTS	I found the authors responsive to the review and the revised manuscript version is a clearer, more succinct version of the original submission
--

REVIEWER	Zainab Samaan McMaster University, Canada
REVIEW RETURNED	23-Mar-2014

GENERAL COMMENTS	The major concern still remains in my opinion is the lack of adjustment for psychopathology. Data on psychopathology are available however the authors are reluctant to include in the current analysis to adjust for this important variable known to influence suicidal behaviour. The second major concerns is the lack of estimation of power to test the study hypothesis. the authors provided details of their sampling methods and although the baseline population sample is large the number of events of the outcome of the study (suicidal behaviour) is small. The third concern remaining for this study is the lack of adjustment for multiple hypotheses testing as mentioned in my previous review. In the PDF version I reviewed the participants' flowchart is missing
---

	the numbers at each level. These numbers should be completed to account for the total sample included.
--	--

VERSION 2 – AUTHOR RESPONSE

Reviewer: 2

Reviewer Name Ingunn Rangul Askeland

Institution and Country Norwegian centre for violence and traumatic stress studies

Norway

Please state any competing interests or state 'None declared': None declared

I found the authors responsive to the review and the revised manuscript version is a clearer, more succinct version of the original submission

Reviewer: 3

Reviewer Name Zainab Samaan

Institution and Country McMaster University, Canada

Please state any competing interests or state 'None declared': none declared

1. The major concern still remains in my opinion is the lack of adjustment for psychopathology. Data on psychopathology are available however the authors are reluctant to include in the current analysis to adjust for this important variable known to influence suicidal behaviour.

We added mental disorders in the final model, please see table 5b. Previously numbered table 5 is now table 5a. This new table (Table 5b) controls for mental disorders (group significance test for mental disorders shown in the last row of the table).

After controlling for mental disorders, the results were largely unchanged, i.e. confirming results as displayed in table 5a. Only one additional variable emerged when controlling for mental disorders, namely sexual abuse was significantly associated with suicidal ideation (Table 5b). We have included this in the results section, see p. 14. We have also included this in the discussion on p. 17.

2. The second major concern is the lack of estimation of power to test the study hypothesis. The authors provided details of their sampling methods and although the baseline population sample is large the number of events of the outcome of the study (suicidal behaviour) is small.

A power calculation based on logistic regression with one continuous predictor variable was conducted, where a 10% prevalence of suicidal behaviour was used and a rounded sample size of 4000. The target odds ratio was set at 2. Based on an N of 4000 (given alpha of 0.05, 2 sided significance), the study was adequately powered (.99) (please see page 7 in the main document).

The POWER Procedure

Likelihood Ratio Chi-Square Test for One Predictor

Fixed Scenario Elements

Method Shieh-O'Brien approximation

Alpha 0.05

Response Probability 0.1

Test Predictor X

Odds Ratio for Test Predictor 2

Unit for Test Pred Odds Ratio 1

Total Sample Size 4000

Total Number of Bins 10

Computed Power

Power

>.999

3. The third concern remaining for this study is the lack of adjustment for multiple hypotheses testing as mentioned in my previous review.

We have added the set of predictor chi square tests to our analyses. Please find attached Table 3-5 with demographics/parent psychopathology added. We controlled for demographics and significant interactions between demographics and intervals, so for the significance test, we included group significance for demographics by themselves, group significance for the interactions between demographics and intervals (only the ones that were included in the model as controls), and both together. Tables 3 and 4: Rows of significance tests were added to these tables in the bottom (rows 15-19) – this is the significance of the multivariate models. The same has been done for tables 5a and 5b repeats controlling for mental disorders, and row 19 includes the group significance test for mental disorders

4. In the PDF version I reviewed the participants' flowchart is missing the numbers at each level.

These numbers should be completed to account for the total sample included.

We added the numbers in person years (please see pdf figure with changes)